



# Response of the IEA Wind 15 MW – WindCrete and Activefloat floating wind turbines to wind and second-order waves

Mohammad Youssef Mahfouz[1], Climent Molins[2], Pau Trubat[2], Sergio Hernández[3], Fernando Vigara[3], Antonio Pegalajar-Jurado[4], Henrik Bredmose[4], and Mohammad Salari[1]

[1]Stuttgart Wind Energy (SWE), University of Stuttgart, Allmandring 5B, 70569 Stuttgart, Germany
[2]UPC-Barcelona-Tech, Campus Nord, Carrer de Jordi Girona, 1, 3, 08034, Barcelona, Catalunya
[3]Esteyco, SA, Menéndez Pidal, 17, 28036 Madrid, Spain
[4]Department of Wind Energy, Technical University of Denmark, Nils Koppels Allé 403, 2800 Kongens Lyngby, Denmark

**Correspondence:** Mohammad Youssef Mahfouz (mahfouz@ifb.uni-stuttgart.de)

**Abstract.** The EU Horizon 2020 project COREWIND has developed two floating platforms for the new International Energy Agency (IEA) Wind 15 MW reference model. One design – "WindCrete" – is a spar floater, and the other – "Activefloat" – is a semi-submersible floater. In this work the design of the floaters is introduced with their aero-hydro-servo-elastic numerical models, and the responses of both floaters in both static and dynamic simulations are verified against the operational and survival design limits. The static displacements and natural frequencies are simulated and discussed. Additionally, the effects of the mean wave drift forces, and difference second order wave forces on the systems' responses are presented. The increase in the turbine's power capacity to 15MW in IEA Wind model, leads to an increase in inertial forces and aerodynamic thrust force when compared to similar floating platforms coupled to the Danish Technical University (DTU) 10MW reference model. The goal of this work is to investigate the floaters responses at different load cases. The results in this paper suggest that at mild wave loads the motion responses of the 15MW Floating Offshore Wind Turbines (FOWT) are dominated by low frequency forces. Therefore, motions are dominated by the wind forces, and second order wave forces rather than the first order wave forces. After verifying and understanding the models' responses, the two 15MW FOWT reference numerical models are publicly available to be used in the research and development of floating wind energy.

## 1 Introduction

Floating Offshore Wind Turbines (FOWTs) will play a key role in the transition towards renewable and sustainable energy systems. In Europe, 80% of the offshore wind energy resources lies in deep water regions (Hundleby and Freeman, 2017). Going into deeper water sites opens the door for bigger turbines with higher energy capacities, and so there is a need of new concept models that can be used for academic research and innovation in the field of FOWTs. Specifically, there is a need for aero-servo-hydro-elastic models of the coupled floater and wind turbine.

COREWIND (COst REduction and increase performance of floating WIND technology) is a Horizon 2020 project aiming to decrease the Levelised Cost Of Energy (LCOE) of FOWTs by 15%, through the optimization of the mooring lines and the power cable. Two FOWTs conceptual designs will be used to validate the innovations presented in COREWIND for mooring





and cable design and optimization. Moreover, wave tank tests as well as wind tunnel tests will be used to validate the models introduced throughout the project period of forty two months. The project includes thirteen participants from both industry and
academics fields.

COREWIND is designing two conceptual floaters for the IEA Wind 15 MW reference turbine model (Gaertner et al., 2020); "WindCrete" is a spar concept floater with a concrete tower, while "Activefloat" is a semi-submersible floater, and a steel tower. OpenFAST v2.1 (NREL, 2019) is used to model the 15 MW FOWTs concepts. The main parameters of the 15MW IEA Wind reference model are shown in Table 1. The tower design and the hub height are adapted for each floater separately, therefore
they are left out in Table 1 since only the Rotor Nacelle Assembly (RNA) parameters are relevant to the FOWTs models presented in this paper. The NREL Reference Open Source Controller (ROSCO) is used for the 15MW IEA Wind reference model (NREL, 2020). ROSCO is a baseline Bladed style controller interface to be used for research purposes. This controller is tuned in order to be adapted to FOWTs.

**Table 1.** IEA Wind 15MW reference turbine parameters

| Parameter | Value |
|-----------|-------|
| Power rating | 15 $MW$ |
| Turbine class | IEC Class 1B |
| Cut in wind speed | 3 $m/s$ |
| Cut-out wind speed | 25 $m/s$ |
| Rotor Diameter | 240 $m$ |
| Rated wind speed | 10.59$m/s$ |
| Blade mass | 65 $t$ |
| Mass of Rotor-Nacelle Assembly (RNA) | 1016 $t$ |

The increase of the power capacity of the turbines to 15MW means an increase in the mass and inertia of both the turbine and
the floater. For the turbine's RNA, the 15MW's RNA has a 50% increase in mass when compared to the DTU 10MW's RNA. For the floaters, WindCrete concrete floater has a 170% increase in mass compared to the 10 MW steel spar floater introduced in Hegseth and Bachynski (2019). Moreover, for the IEA Wind 15MW model the rated thrust is increased by 87% compared to the DTU 10 MW reference model. The effects of the increase of aerodynamic thrust force and the increase of mass and inertia of the FOWT on the floater's response are shown in this paper.

In this paper, a short introduction to controller, hydrodnamics and mooring numerical models in OpenFAST is given in section 2. In sections 3, and 4, the design parameters of both FOWTs designs are presented, with an emphasis on the changes done in the OpenFAST model to transform the 15MW fixed bottom offshore model (Gaertner et al., 2020) into a FOWT model. Load cases used to verify the models implementation in OpenFAST, and to show the effects of wind and second order waves forces on the system's response are introduced in section 5. The verification's outputs are presented in section 6. In this section,
the natural frequencies of the models, as well as the static equilibrium of the floaters are presented. Additionally, the tuned controller's performance is initially checked using step wind, in the absence of waves. Moreover, the effect of the increase





of the FOWTs mass and inertia can be clearly seen in regular waves simulations in the absence of wind. Second order wave effects are shown using irregular wave simulations. Finally, the dynamic system's response to turbulent wind and irregular waves is shown along with the system's response to extreme 50 years wind and waves. The verification process focuses on
the platforms' responses to different excitation forces, analyzing which forces dominate the platforms' motions in different Degrees Of Freedom (DOFs).

## 2    Numerical Modelling using OpenFAST

OpenFAST is an aero-hydro-servo-elastic tool, developed by NREL, to model offshore (fixed bottom and floating) as well as onshore wind turbines (Jonkman, 2007). The tool uses a combination of modal and multibody dynamics formulation.
OpenFAST models the blades, and the tower as elastic beams while the platform is modelled as a rigid body. The coordinate system used throughout this paper is identical to the reference coordinate system defined in OpenFAST. The right handed coordinate system has a positive x-axis pointing downwind, while the positive z-axis is pointing upwards and the global reference frame origin is at the mean sea water level. The aerodynamic forces are modelled in OpenFAST using Blade Element Momentum (BEM) theory with Aerodyn. The hydrodynamic forces are calculated using both potential flow theory, and strip
theory with Hydrodyn (Jonkman et al., 2015). Mooring lines forces are calculated through Moordyn (Hall, 2017). The forces from Aerodyn, Hydrodyn, and Moordyn are coupled to Elastodyn where the equations of motions of the coupled system are solved.

### 2.1    Controller Design

The ROSCO controller is adopted and re-tuned in this paper. Below the rated wind speed, the ROSO controller includes a
Proportional Integral (PI) controller for generator torque control. The below rated PI controller adjusts the generator speed to follow the optimal tip-speed ratio for harvesting the maximum electrical power. In our models, this controller is used with minor re-tuning. For overrated wind speeds the ROSCO controller uses a PI collective pitch controller to regulate the generator speed at its rated value while the generator torque is also kept constant at rated value (Mulders and Van Wingerden, 2018). Major tuning is done to the above rated wind speed controller due to the unfavourable couplings between tower motion and
pitch controller. This coupling arises when the wind turbine is installed on a floating platform. This is mainly because the lowest natural frequencies in FOWT, which are for surge and pitch motions of platform, are much smaller than those of fixed-bottom platforms, which are usually for tower fore-aft and lateral bending. These low natural frequencies put some limitations on the bandwidth of the pitch controller. For example in Larsen and Hanson (2007), it has been shown that applying a controller, which has been tuned for a wind turbine installed onshore, on the same turbine installed on a floating platform can lead to
instability. A straightforward approach to deal with this challenge is to de-tune the controller to not let the undesired coupling between tower motion and pitch controller lead to instability.

The PI collective pitch controller for overrated has been designed using the Ziegler-Nichols approach (Ziegler et al., 1942). For a specific overrated wind speed, the PI gains are calculated and the proportion between original ROSCO PI parameters





and the calculated ones through Ziegler-Nichols for this specific wind speed are then used to scale the controller parameters

for all above-rated wind speeds. This re-tuning of the controller has been carried out for the WindCrete FOWT and has been successfully applied on the ActiveFloat case without the need to update the parameters. Step wind simulations are carried out to ensure the controller performs as expected and can be seen later in Figure 5.

## 2.2 Modelling of Hydrodynamics

Hydrodynamic forces are modelled in Hydrodyn using potential flow theory and strip theory. The potential flow theory forces

act on the rigid floater at mean sea water level. The potential flow solver ANSYS-AQWA (ANSYS, 2015) is used to solve the potential flow theory and provide the added mass $\boldsymbol{A}(\omega)$, radiation damping $\boldsymbol{B}(\omega)$, which are functions of the wave frequency. First order wave forces $\boldsymbol{X}(\omega)$, and difference frequency second order wave forces $\boldsymbol{X}^-(\omega_m,\omega_n)$ are calculated using the potential flow solver and are functions of wave direction as well as wave frequency. The frequency domain representation of the hydrodynamic loading is shown in equation 1, where $\boldsymbol{C}$ is the hydrostatic stiffness, $\boldsymbol{x}$ is the a vector of the six DOFs

of the platform, and $\boldsymbol{F}$ is the first and second order wave forces acting on the platform. OpenFAST uses Cummins equation (Cummins, 1962) to couple the frequency-dependent hydrodynamic properties to the time-domain solution (Jonkman, 2009).

$$(-\omega^2(\boldsymbol{M}+\boldsymbol{A})+i\omega\boldsymbol{B}+\boldsymbol{C})\boldsymbol{x}(\omega)=\boldsymbol{F}(\omega) \tag{1}$$

The second order wave loads are proportional to the square of the wave amplitude and they have frequencies of the sum and difference frequencies of the linear wave spectrum. Although second order forces have lower amplitudes than the first order

ones, they can excite the natural frequencies of the floater especially the lower ones such as the surge natural frequencies. This can lead to higher fatigue loads in the FOWT system (Duarte et al., 2014).The summation of the diagonal components of the Quadratic Transfer Function (QTF) represents the mean drift force acting on the platform. Throughout this paper, the second order forces are applied to the floaters using the difference QTF as the sum QTF is not relevant for our system because of its low natural frequencies.

## 100 2.3 Modelling Mooring Lines

Moordyn is used to model the mooring lines in OpenFAST (Hall, 2017). Moordyn is a dynamic lumped mass model, the position and the velocity of the platform are provided to Moordyn at every coupling time step. Moordyn calculates the overall forces acting on the platform in the six DOFs and provides a force vector back to OpenFAST. In both models, catenary mooring systems are used for station keeping. The mooring designs presented here are preliminary designs and are not verified against

Ultimate Limit State (ULS), Accidental Limit State (ALS), and Fatigue Limit State (FLS). Optimised mooring designs for the floaters will be created later on during the COREWIND project.



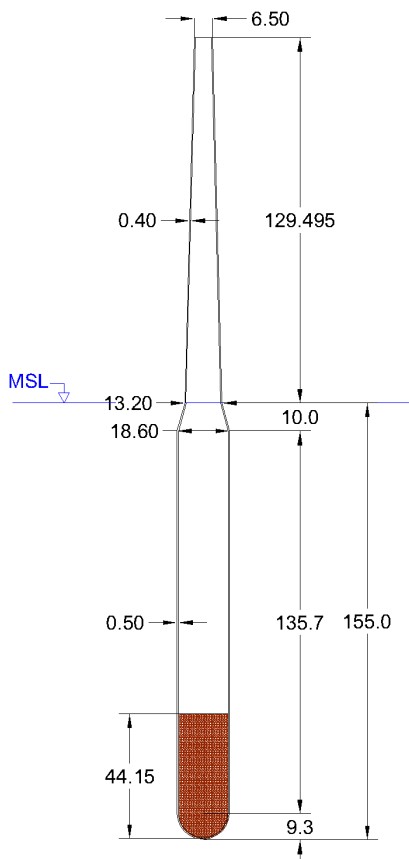

**Figure 1.** WindCrete geometry (units in meter)

## 3 WindCrete

WindCrete (Campos et al., 2016) concept by Universitat Politècnica de Catalunya (UPC), is a monolithic spar design, with a draft of 155m. The wind turbine tower and the spar are one member made of concrete with no connecting joints between them, this increases the durability of WindCrete by removing weak points in the structure. A ballast is added at the bottom of the floater (coloured area in Figure 1) to increase the hydrostatic stiffness in the pitch direction. The submerged spar section is made of three parts:

– A semi-sphere at the bottom to distribute the hydrostatic pressure over its surface. However, this comes with the draw back of reducing the heave axial damping. However, this does not represent a significant problem because spar platforms have low wave excitation forces in the heave degree of freedom (DOF) due to their deep draft.

– A straight cylinder which provides the buoyancy required, as well as carrying the ballast mass.



- A truncated cone section, which connects the tower to the floater.

The tower is conical shape, with a base diameter larger than the fixed bottom offshore reference model, to withstand the higher tower base moments produced by the tower top masses due to the pitch and roll motions of the floater. The hub height of the model is 135m above mean sea level. The mass and inertia parameters of the submerged substructure including the ballast are shown in Table 2.

**Table 2.** WindCrete substructure parameters

| Parameter | Value |
|---|---|
| Mass including ballast | 3.655e+07 $kg$ |
| Vertical Center of Gravity (VCG) | -113.08 $m$ |
| $I_{xx}$ about CG | 5.590e+10 $kg.m^2$ |
| $I_{yy}$ about CG | 5.590e+10 $kg.m^2$ |
| $I_{zz}$ about CG | 1.828e+09 $kg.m^2$ |

### 3.1 Hydrostatics of WindCrete 15MW

For spar floaters, the Center of Gravity (CG) of the over all system must lie below the center of buoyancy (CB), in order to provide pitch and roll stability for the turbine. WindCrete was designed, following the approach presented in Matha et al. (2015), such that the static mean pitch angle at rated thrust is equal to $3.2°$, and that the tower base can withstand the fatigue and ultimate loads due to the pitch and roll motions. The hydrostatic parameters of the overall system (WindCrete + RNA) can be found in Table 3. The roll and pitch hydrostatic stiffnesses in Table 3 are purely hydrostatic and they become positive when the contribution of the center of gravity is added.

**Table 3.** WindCrete + RNA parameters

| Parameter | Value |
|---|---|
| Overall mass | 3.9805e+07 $kg$ |
| Center of Gravity (CG) | -98.41 $m$ |
| Center of Buoyancy (CB) | -77.29 $m$ |
| $I_{xx}$ at CG | 1.5536e+11 $kg.m^2$ |
| $I_{yy}$ at CG | 1.5536e+11 $kg.m^2$ |
| $I_{zz}$ at CG | 1.9025e+09 $kg.m^2$ |
| Displaced water volume | 4.054e+04 $m^3$ |
| Heave stiffness $C_{33}$ | 1.3746e+06 $N/m$ |
| Roll hydrostatic stiffness $C_{44}$ at sea water level | -3.1463e+10 $N \cdot m/rad$ |
| Pitch hydrostatic stiffness $C_{55}$ at sea water level | -3.1463e+10 $N \cdot m/rad$ |





## 3.2 Hydrodynamics of WindCrete 15MW

The detailed potential flow solution for the added mass, radiation damping, first and second order wave excitation forces of WindCrete is presented in Mahfouz et al. (2020b). In order to include viscous effects to the model, the strip theory in Hydrodyn applies the Morison equation on the elements defined in the model. In the WindCrete model, two drag coefficients are defined for the transverse and the axial direction. The transverse drag is equal all over the submerged section of WindCrete with a value of 0.7 (Campos et al., 2015). The axial drag is applied at the hemisphere geometry at the bottom of WindCrete. The axial drag

coefficient is equal to 0.2 following (Hoerner, 1965). The effects of marine growth are not considered in this work.

## 3.3 Mooring Lines

Three catenary delta shape mooring lines are used for station keeping of the WindCrete floater. The mooring lines provide stiffness for surge, sway, and yaw DOFs. The yaw stiffness is a critical parameter for spar floaters and needs to be big enough to ensure that the yaw natural frequency is much smaller than the roll natural frequency in order to avoid aerodynamic yaw-roll

coupling (Haslum et al., 2020).

The mooring line system consists of three symmetric catenary mooring lines. Each line is composed of a single chain at the bottom with a length of $565m$, connected to a delta shape connection with a length of $50m$. The three lines are made of one type chain with a diameter of $160mm$, dry weight of $561.25kg/m$, and axial stiffness of $2.304E + 09N$. The geometry of the lines is presented in Table 4.

**Table 4.** WindCrete mooring system's fairlead and anchors positions

| Line | Anchor coordinates [m] | | | Fairlead coordinates[m] | | |
|---|---|---|---|---|---|---|
| | X | Y | Z | X | Y | Z |
| 1 | -600 | 0.0 | -200 | -4.65 | 8.05 | -90.0 |
| | | | | -4.65 | -8.05 | -90.0 |
| 2 | 300 | -519.61 | -200 | -4.65 | -8.05 | -90.0 |
| | | | | 9.3 | 0.0 | -90.0 |
| 3 | 300 | 519.61 | -200 | 9.3 | 0.0 | -90.0 |
| | | | | -4.65 | 8.05 | -90.0 |

## 4 Activefloat


The Activefloat concept developed for this paper by Esteyco is a semi-submersible floater concept. The floater is made of concrete to increase its durability. The structure of the floater consists of three external columns, a central column, and three pontoons connecting the external columns to the central one. The tower is a steel structure connected to the floater at the



central column. The draft of the platform is $26.5m$. An active ballast system keeps the static mean pitch at zero degrees for all
operational wind speeds. The main parameters of the floater, shown in Figure 2, can be described as follows:

– Three external columns form an equilateral triangle. These columns provide the pitch and roll stability for the platform.
  At the bottom of each column lies a heave plate to damp the heave motion of the platform. The external columns are
  hollow and partially filled with water, the water level in each column is controlled by the active ballast system.

– The central column has a conical shape and the tower is connected to the platform through the central column. The
central column is totally left dry, to contain the machinery required for example for the active ballast system.

– The pontoons are connecting the three external to the central column, and they are fully filled with water all the time.

The tower is a conical shape steel tower, similar to WindCrete the hub height is at $135m$ above sea water level.

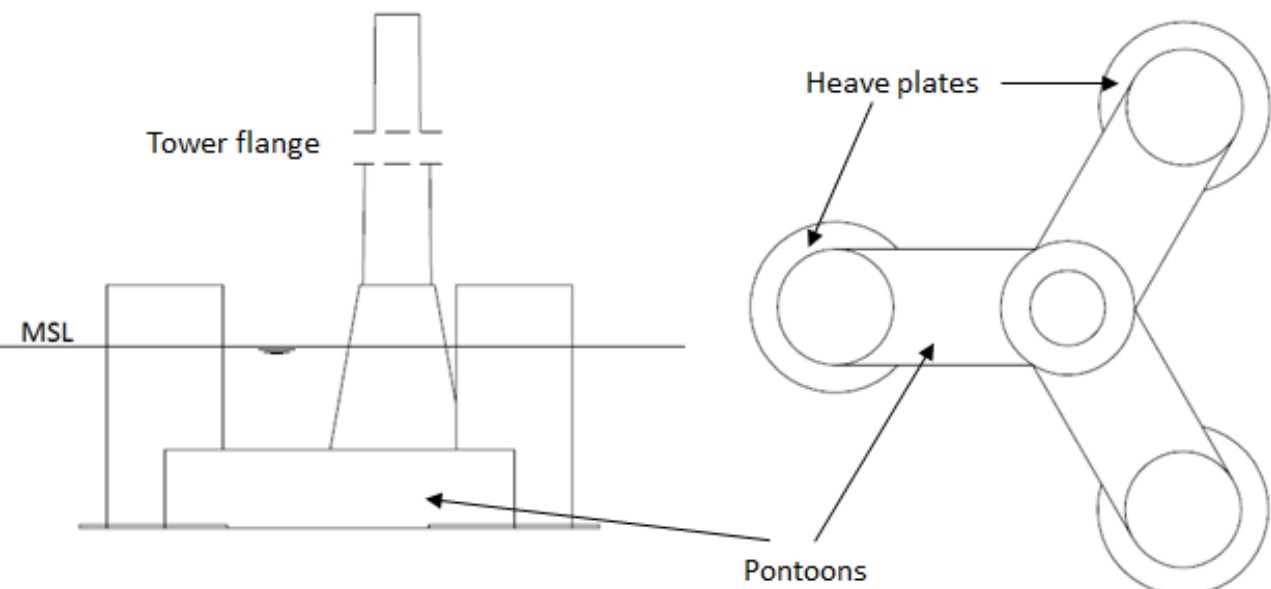

**Figure 2.** Activefloat geometry (side view on the left and top view on the right)

### 4.1  Hydrostatics of Activefloat 15MW

The mass and inertia properties of the Activefloat floater without the tower and the RNA is shown in Table 5. In the OpenFAST
model, $100t$ are added to the platform mass to account for all the machinery included inside the tower. The $100t$ of extra mass
are assumed to be at the tower's CG, the total mass of the platform and its CG are adjusted accordingly.

The active ballast is controlled by a pump arrangement exchanging water between the external columns according to the
mean thrust force acting on the wind turbine's rotor. The ballast mass is equally divided between the three external column





**Table 5.** Activefloat substructure parameters

| Parameter | Value |
|---|---|
| Mass including ballast | 3.4387e+07 $kg$ |
| Vertical Center of Gravity (VCG) | -17.529 $m$ |
| $I_{xx}$ about CG | 1.57e+10 $kg.m^2$ |
| $I_{yy}$ about CG | 1.57e+10 $kg.m^2$ |
| $I_{zz}$ about CG | 2.58e+10 $kg.m^2$ |

whenever the active ballast is deactivated. In this work the active ballast is deactivated during idling or parked conditions. The active ballast system's schedule for Normal Turbulence wind Model (NTM) and Extreme Turbulence wind Model (ETM) is presented in Table 6.

**Table 6.** Activefloat active ballast schedule

| Turbulence model | Wind speed [m/s] | Overall floater's CG [m] | | |
|---|---|---|---|---|
| | | X | Y | Z |
| Extreme turbulence | 8 | -0.379 | | |
| | 10.5 | -0.522 | | |
| | 16 | -0.259 | | |
| | 20 | -0.196 | | |
| | 25 | -0.181 | 0 | -17.59 |
| Normal turbulence | 8 | -0.365 | | |
| | 10.5 | -0.594 | | |
| | 16 | -0.272 | | |
| | 20 | -0.207 | | |
| | 25 | -0.188 | | |

## 4.2 Hydrodynamics of Activefloat 15MW

The detailed potential flow solution for the added mass, radiation damping, first and second order excitation forces of Active-float is presented in Mahfouz et al. (2020b). In order to include the viscous effects, Activefloat is modelled in Hydrodyn as a
number of Morison elements. However, Hydrodyn only allows modelling of cylindrical elements as Morison elements. Hence, Activefloat pontoons can not be directly modelled in Hydrodyn with their rectangular faces. To overcome this limitation the approach presented in Pegalajar-Jurado et al. (2018) is adopted. A detailed description of the Morison elements implemented in Hydrodyn can be found in Mahfouz et al. (2020b). The effects of marine growth are not included in this work.





### 4.3 Mooring Lines

The mooring system used for station keeping of Activefloat is made of three symmetric catenary mooring lines, where a line is attached to each platform arm. The lines are made of chain of weight $561.25kg/m$, axial stiffness of $2.304e+09N$, a diameter of $0.16m$, and length of $614m$. The mooring lines' design ensure only horizontal loading on the anchors, and that the maximum excursion in surge at rated thrust value is below $20m$. The geometry of the lines is presented in Table 7.

**Table 7.** Activefloat's mooring lines system

| Line | Anchor coordinates [m] | | | Fairlead coordinates [m] | | |
|------|------|------|------|------|------|------|
|      | X    | Y    | Z    | X    | Y    | Z    |
| 1    | -600 | 0.0  | -200 | -42.5 | 0.0  | -15 |
| 2    | 300  | -519.6152 | -200 | 21.25 | -36.806 | -15 |
| 3    | 300  | 519.6152 | -200 | 21.25 | 36.806 | -15 |

### 5   Load Cases

A summary of the load cases used to validate the models in OpenFAST is presented in Table 8. The load cases were selected to identify the main characteristics of the FOWTs under static and dynamic loads, and to find out the effects of the second order wave forces and the increase of mass of the system. First, the static equilibrium of the floaters in the absence of wind and wave forces is calculated. Afterwards, the natural frequencies are calculated using free decay tests. The natural frequencies of the tower in fore-aft and side to side is also calculated for both platforms, to ensure that the new towers' designs natural

frequencies lay outside of the 3P frequency region of the rotor. The controller response is checked using step wind simulations load case 7 in Table 8. The wind is increased from 3m/s to 25m/s with a 1m/s step, then decreased again to 3m/s. The step time is 200s for every wind speed.

In order to check the effect of second order wave excitation forces on the platform, the response of the platform to first and second order excitation forces, of regular and irregular waves, is checked in load cases 8,9,10, and 11. Moreover, the

dynamic responses of the FOWTs during operation and extreme conditions are investigated by a number of simulations with turbulent wind fields and irregular waves. Operation of the turbine in a wind field with Extreme Turbulence Model (ETM), and Normal Sea State (NSS) with second order wave forces is checked in load case 13 in Table 8. Also simulations with a Normal Turbulence Model (NTM) wind field at rated wind speed and Extreme Sea State (ESS) are carried out. Finally, the responses of the FOWTs to 50 years Extreme Wind Model (EWM), and ESS waves are checked. All simulations are carried out with

wind and wave aligned to each other. The environmental conditions of the Gran Canaria Island site presented in Vigara et al. (2020) are used in all of the load cases shown in Table 8. Pierson-Moskwitz spectrum is used for irregular waves generation, and turbulence class C is used for the turbulent wind fields creation. The turbulent wind fields are created using the Kaimal turbulence model following the International Electrotechnical Commission (IEC) standard for offshore wind turbines (IEC,





2019). For load cases 10, 11, 12, 13, and 14 in Table 8, the simulations were run for 5400s but the first 1800s were neglected

to remove any transient effects.

**Table 8.** Load cases used in OpenFAST for models verification

| Load Case | Description | Duration [s] | Wind | Wave | Turbine |
|---|---|---|---|---|---|
| 1 | Static equilibrium | 1500 | - | - | Idling |
| 2 | Surge decay | 1500 | - | - | Idling |
| 3 | Heave decay | 1500 | - | - | Idling |
| 4 | Pitch decay | 1500 | - | - | Idling |
| 5 | Yaw decay | 1500 | - | - | Idling |
| 6 | Tower decay | 1500 | - | - | Idling |
| 7 | Step wind | 9200 | uniform wind, 3-25 m/s | - | Operating |
| 8 | Regular waves | 3000 | - | H= 2m, T= 6s | Idling |
| 9 | Regular waves | 3000 | - | H=2m, T=6s, QTF | Idling |
| 10 | Extreme irregular waves | 5400 | - | $H_s$=5.11 m, $T_p$= 11s | Idling |
| 11 | Extreme irregular waves | 5400 | - | $H_s$=5.11m, $T_p$=11s, QTF | Idling |
| 12 | Opertion at NTM wind and ESS | 5400 | NTM, 10.5 m/s | ESS, $H_s$= 5.11m, $T_p$= 11s, QTF | Operating |
| 13 | Operation at ETM wind and NSS | 5400 | ETM, 10.5 m/s | NSS, $H_s$= 2m, $T_p$= 6s, QTF | Operating |
| 14 | 50 years extreme wind and wave | 5400 | EWM 50, 28.35 m/s | ESS, $H_s$= 5.11m, $T_p$= 11s, QTF | Idling |

## 6  Floaters Responses

Verification of the floaters responses to the load cases introduced in Table 8 are presented. In all load cases the waves are coming from zero degrees heading, therefore we focus on the platforms' responses in surge, heave and pitch DOFs. We emphasize on the response of the spar and the semi-submersible floaters' to difference second order QTF, and how the responses are different

for the spar and semi-submersible. The spar floater is known to have better responses in heave than semi-submersible floater, on the other hand it is more sensitive to the pitch, roll, and yaw motions due to its small water plane area (Roddier et al., 2010). The effect of the larger mass, inertia and aerodynamic thrust force is investigated using coupled wind and wave simulations. The effect of the second order drift forces is determined through regular waves simulations with and without second order forcing in the absence of wind. The effect of the second order forcing at low frequencies is shown by simulating irregular

waves with and without difference second order wave excitation forces.

### 6.1  Static Equilibrium

Table 9 shows the static position of both floaters, to check the balance between the hydrostatic forces, the mooring forces and the gravitational forces in the absence of wind and waves. The negative pitch comes from the big overhang distance of





the RNA, where the CG of the RNA is located $-7.01925m$ in x direction. In Activefloat, the pitch angle is higher due to the
asymetric mass distribution of the mooring lines masses around the y-axis. The surge offset from the zero position comes from
the mooring lines tensions in the x direction in both floaters.

**Table 9.** Floater's static equilibrium

|  | Surge [m] | Heave [m] | Pitch [deg] |
|---|---|---|---|
| WindCrete | -1.01 | -0.16 | -0.64 |
| Activefloat | 0.052 | 0.025 | -1.799 |

## 6.2 Free Decay Tests

In order to calculate the natural frequency of the FOWTs at a specific DOF, the FOWT was excited in this DOF, and left to
oscillate freely. For heave, roll, and pitch the natural frequency depends on the mass of the overall system and the hydrostatic
stiffness. For surge, sway and yaw, the hydrostatic stiffness is zero and mooring lines provide stiffness for the system. Therefore,
for surge, sway, and yaw, the natural frequency of the system depends on the mooring lines design. For spar floaters, to avoid
roll yaw coupling, the natural frequency in yaw must be much higher than the roll natural frequency (Haslum et al., 2020). This
constraint is taken into account while designing the mooring lines for WindCrete.

The natural frequencies of both floater can be seen in Table 10. The surge and pitch free decay time series can be seen in
Figure 3. The surge decay includes not only one frequency, but a combination of different frequencies because it is measured
at the mean sea level and not at the CG of the FOWT system. The surge and pitch free decay time series can be seen in
Figure 4. The towers' fore-aft and side to side natural frequencies (see Table 10 are always higher than the rotor 3P frequencies
calculated in Gaertner et al. (2020) to be between 0.25 and 0.38 Hz.

**Table 10.** Floaters' and towers natural frequencies

|  | Surge | Heave | Pitch | Yaw | Tower |
|---|---|---|---|---|---|
| WindCrete's natural frequency [Hz] | 0.01221 | 0.03052 | 0.02441 | 0.09155 | 0.5 |
| Activefloat's natural frequency [Hz] | 0.0061 | 0.05493 | 0.0305 | 0.01221 | 0.44 |

## 6.3 Step Wind

In order to check the controller's performance, step wind simulation load case 7 in Table 8, was done on both FOWTs, and
the responses are checked. The steady wind field used increases from 3m/s to 25m/s and then decreases back to 3m/s with
a step duration of 200s. The responses for WindCrete, shown in Figure 5, and Activefloat shown in Appendix A Figure A1
demonstrate that the controller does not introduce negative damping, and there is no platform pitch instability. For WindCrete
spar platform, there are higher fluctuation in the heave direction due to the low heave damping for the small heave amplitude



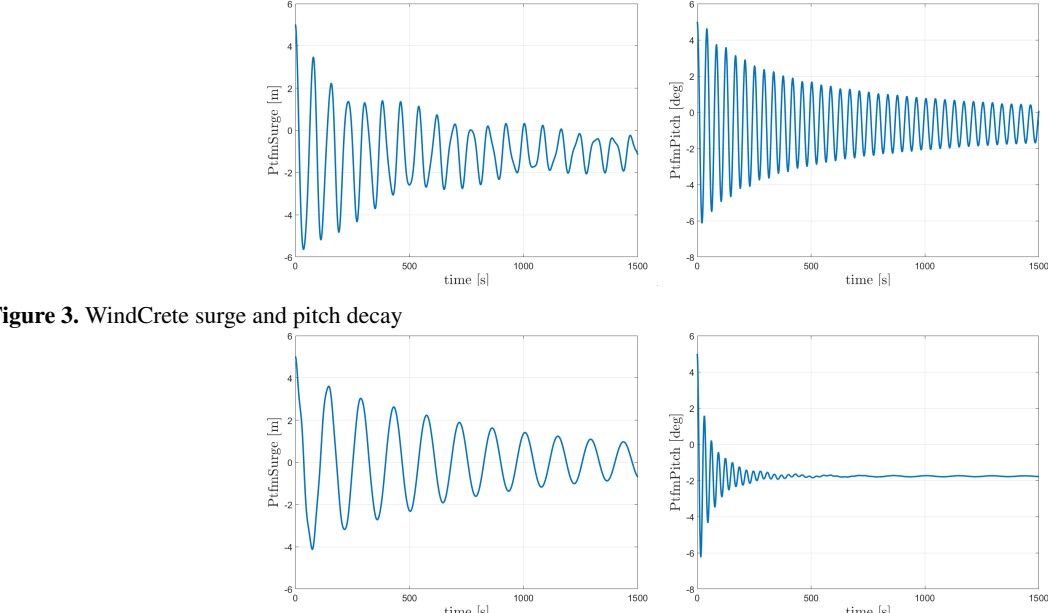

**Figure 3.** WindCrete surge and pitch decay

**Figure 4.** Activefloat surge and pitch decay

fluctuation. Moreover, the change in the floaters' pitch angle causes a change in the vertical forces acting on the FOWT. The 200s between each steady wind step is not enough for the platforms' motions to be completely damped.

### 6.4 Regular waves

Regular wave load cases 8, and 9 in Table 8 are shown in Figures 6, and 7. The figure shows the response of WindCrete and Activefloat to regular waves (H=2m, T=6s) with and without including the difference second order wave forcing. For regular

wave simulations, the difference second order wave forces represent only the mean wave drift force, which is a constant force over time (Pereyra et al., 2016).The frequency response of WindCrete at the natural frequency of the floater shows that the transient response is still seen after 3000s. The limited effect of the wave forces on the floaters response is due to the high inertia of the system as well as the mild waves at the Gran Canaria site.

In the absence of mean drift forces, the floaters' mean static response is equal to the static equilibrium positions shown in

Table 9. For Activefloat, the drift forces change the static mean surge from $0.2m$ to $1.3m$, while the pitch and heave responses are not affected. While adding mean drift forces, changes the mean static response in surge and pitch for WindCrete, but has no effect in heave. In WindCrete, the mean drift moment around the y-axis is equal to $1.8MNm$, causing the change of mean static pitch from $-0.64°$ to $2.6°$ shown in Figure 6. Since the results presented in Figure 6 are shown at mean sea level, the increase in static mean surge is not due to the mean drift forces but due to the increase in the static mean pitch of the platform.

In order to decouple the surge response from the pitch response in WindCrete, the results at CG at $-98.41m$ can be seen in Figure 8.

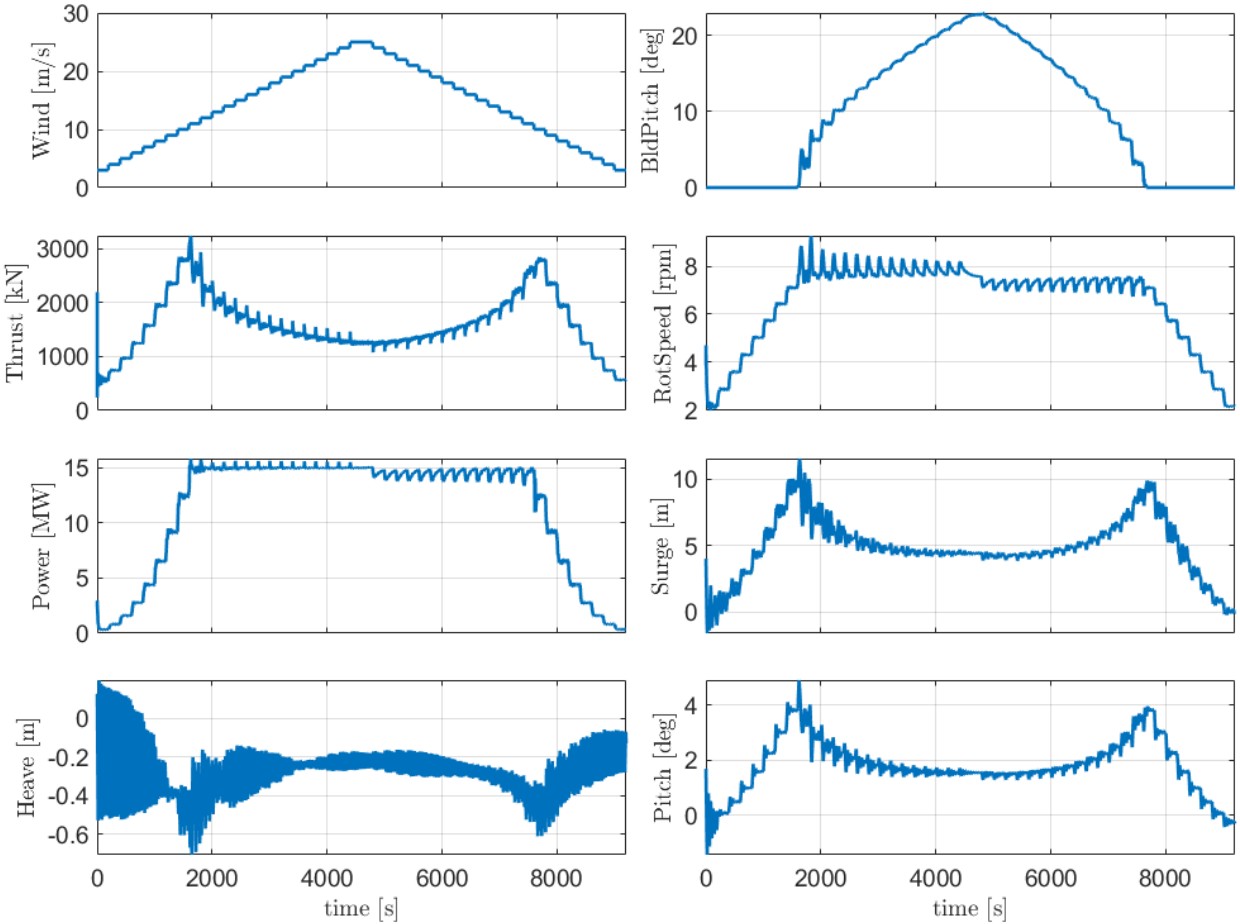

**Figure 5.** WindCrete response to step wind in absence of waves

## 6.5 Irregular waves

Extreme irregular waves simulations ($H_s$= 5.11m, $T_p$=11s) from load cases 10, and 11 in Table 8 are shown in Figures 9, and
10. The second order wave forcing is applied using differentce frequency QTF matrix. The simulation was done for 5400s and
the first 1800s were eliminated to make sure that all the transient responses do not affect the responses shown in our results.
The results show significant resonance effect at low frequency due to the second order wave loads. This low frequency response
is highly affected by the Morison drag coefficients model applied in Hydrodyn (Mahfouz et al., 2020a). Therefore, it cannot
be concluded that the numerical models fit the physical models without conducting a wave tank test data to verify the models
assumptions. The resonance due to second order wave forces is seen for both platforms except for the heave DOF in Activefloat
as the heave motion for semi-submersible floaters is mostly dominated by linear wave forces due to their small draft.





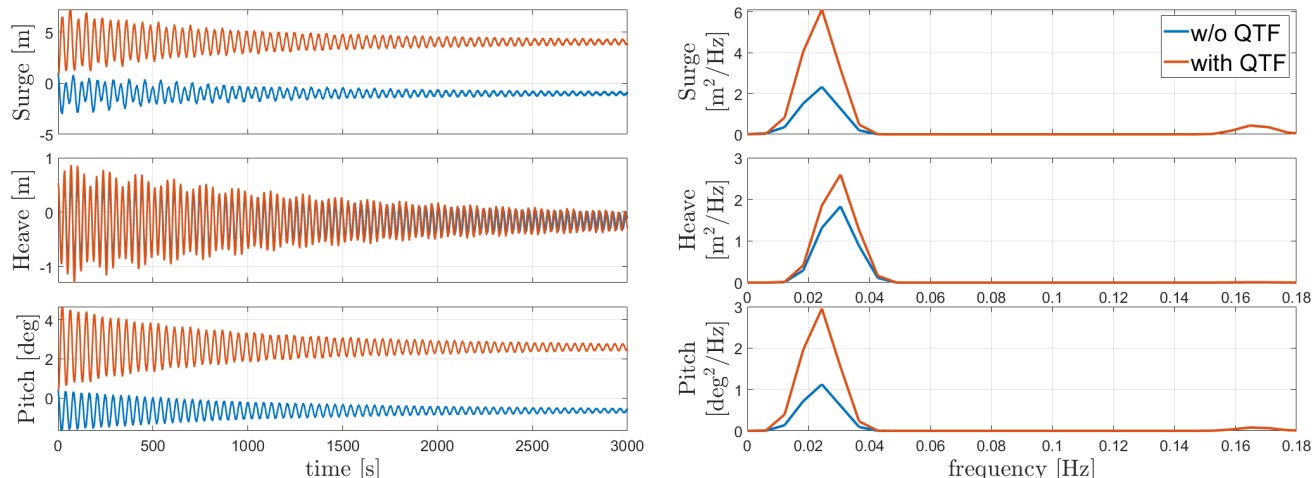

**Figure 6.** WindCrete's response to regular waves without and with second order forces

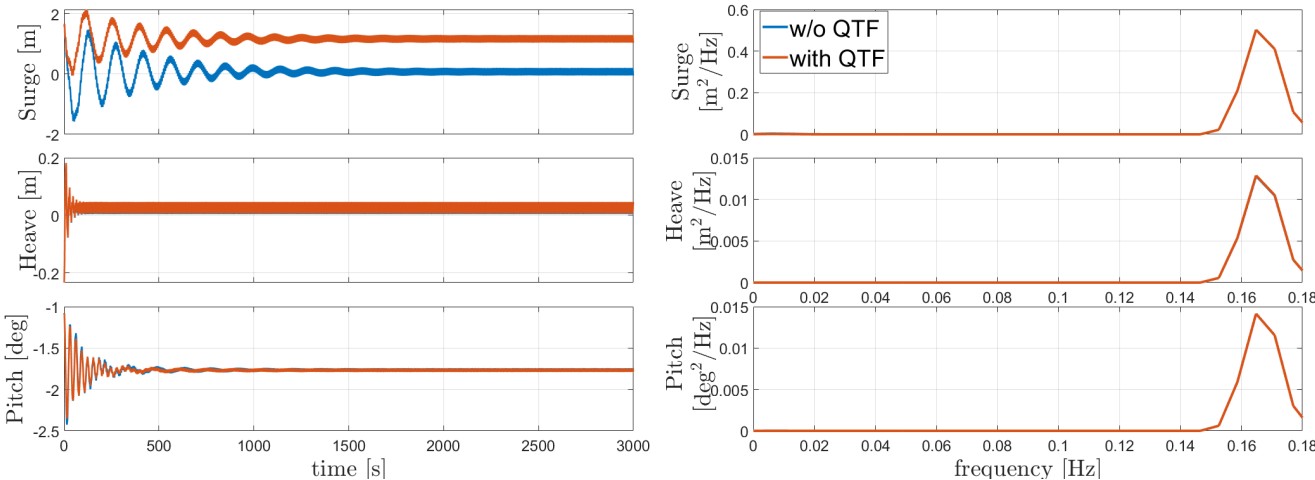

**Figure 7.** Activefloat's response to regular waves without and with second order forces

### 6.6 Operation at NTM wind and ESS

Load case 12 in Table 8, is similar to load cases 10, and 11 except that now a NTM wind field at 10.5m/s is added to the simulation inputs. Moreover, the Activefloat active ballast system is now activated to keep the mean static pitch of the platform around zero. The results of load case 12 with and without second order waves forces, is shown in Figures 11, and 12 for both WindCrete and Activefloat respectively. The figures show that including the second order forces in the presence of wind has a very limited effect on the floaters response, similar to what was shown by Coulling et al. (2013). For WindCrete, the response of surge, heave and pitch DOFs is dominated by their own natural frequencies. The responses at low frequencies are due to the wind forces while waves forces have a very small effect on the response. In Figure 11, the frequency response shows a surge, pitch coupling.






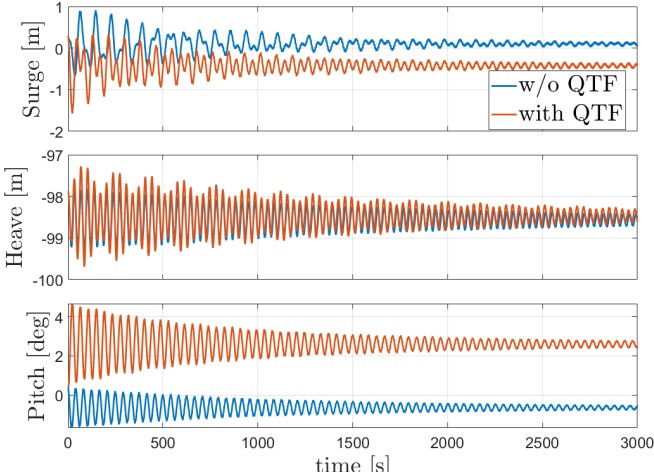

**Figure 8.** WindCrete response to regular waves at CG

In Figures 12 and 10 the Activefloat response in heave DOF is almost identical with and without the NTM wind field. The heave response for Activefloat is dominated by the wave forces due to the small draft of the semi-submersible floater. The frequency responses in surge and pitch shown in Figure 12 are at low frequencies around the platform's natural frequencies and dominated by wind forces.

## 6.7 Operation at ETM and NSS

The FOWTs responses to ETM wind field at 10.5m/s and NSS (irregular waves with $H_s$=2m, and $T_p$=6s) are shown in Figures 13, and 14. The simulations were done for 5400s and the first 1800s were removed to remove transient effects. The response of WindCrete is dominated by the surge, heave and pitch natural frequencies mostly excited by the wind forces, while the wave forcing has minimum effect on the spar's response. Similarly, the response of Activefloat is dominated by low frequency forcing, mainly excited by the wind. In the heave response, the wave forcing can be seen as a small peak around the wave frequency. The mean platform pitch is kept around zero by the active ballast system. The Activefloat's heave response is no longer dominated by the wave forcing for the NSS, because of the mild conditions for our site. However, the heave excitation due to wave forces can still be seen in the frequency response of the system.

## 6.8 Operation at EWM 50 years and ESS

The FOWTs' responses to the site specific EWM 50 years of 28.35m/s and ESS of $H_s = 5.11m$ and $T_p = 11s$ are shown in Figure 15, and Figure 16. The turbine is idling with the blades pitched to $90°$ during both simulations. WindCrete response is dominated by the natural frequency of the floater in surge, heave and pitch. Similarly for Activefloat, the surge and pitch motions are dominated by their natural frequencies. However, the heave response is dominated by the wave forcing frequency. The surge, pitch coupling can be seen for both floaters. The low frequency response is caused by the second order wave forces





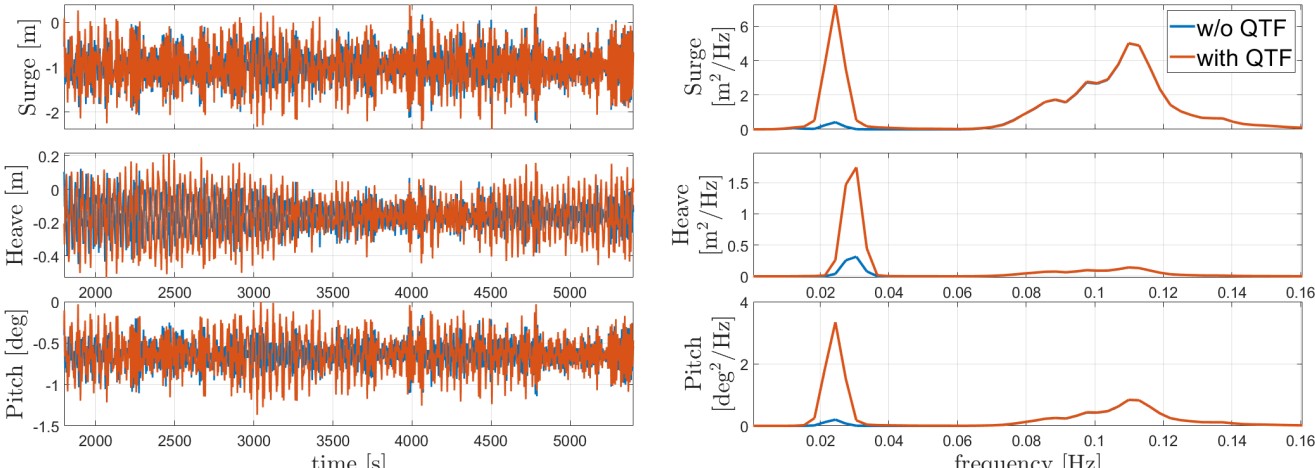

**Figure 9.** WindCrete response to irregular waves in absence of wind

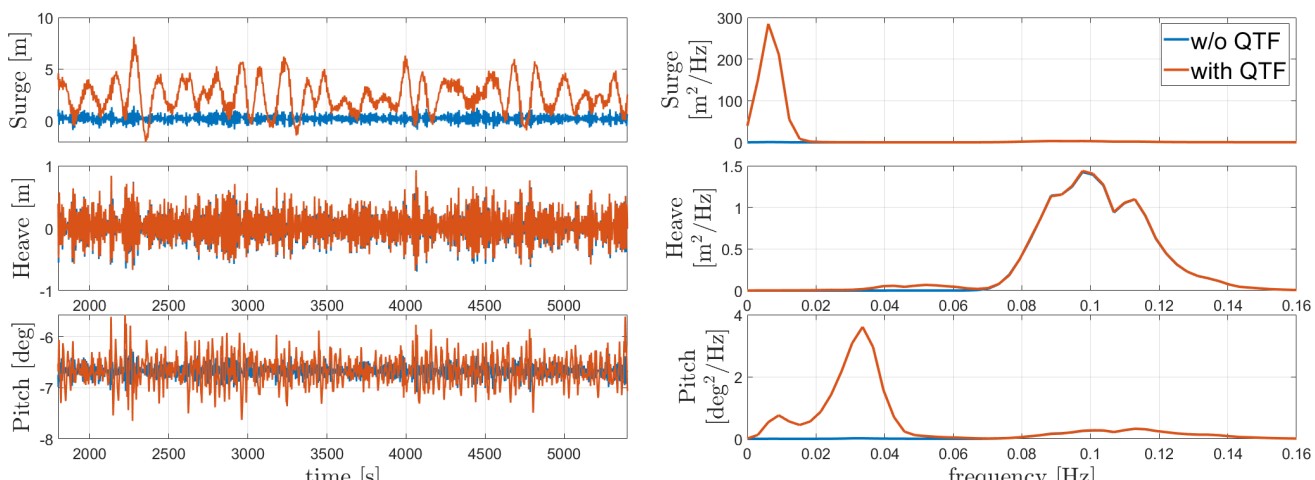

**Figure 10.** Activefloat response to irregular waves in absence of wind

which are more dominant due to the small thrust forces acting on the rotor as the blades are pitched out of the wind. The drift

forces effects can be clearly seen in WindCrete's response (Figure 15), where the platform pitch is excited by the drift forces.

## 7   Conclusion

This paper presents the WindCrete spar OpenFAST model, and the Activefloat semi-submersible OpenFAST model. The floaters were designed within the Horizon 2020 project COREWIND, and were coupled to the IEA Wind 15MW reference wind turbine. The paper introduced the design parameters of the FOWT models with an emphasis on the changes required to

couple the fixed bottom offshore OpenFAST model of the 15MW to the floating platforms. First, the tower was redesigned in





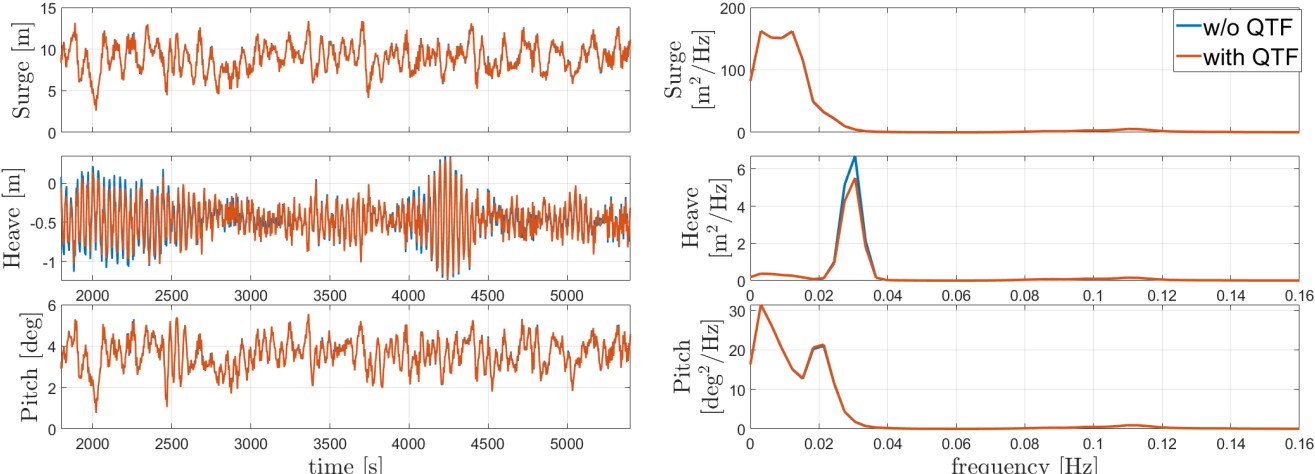

**Figure 11.** WindCrete response to NTM wind and ESS

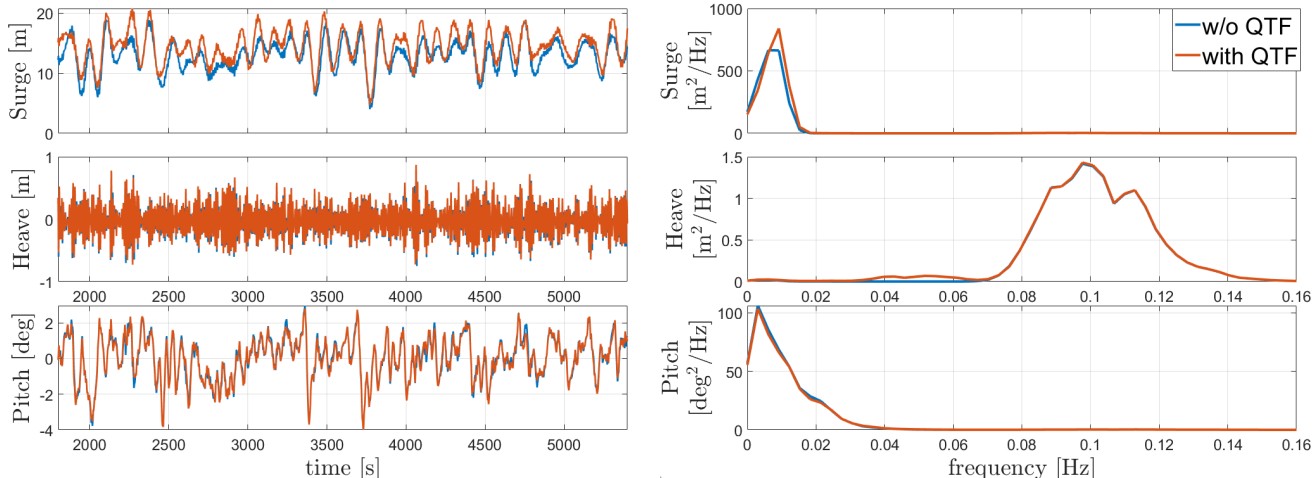

**Figure 12.** Activefloat response to NTM wind and ESS

order to withstand the higher loads at the tower's base. Then, the controller was tuned to avoid negative damping and hence prevent platform pitch instability. Additionally, the hydrodynamics models in Hydrodyn using the potential flow solution, and the strip theory solution to include viscous drag were implemented. Finally, The mooring line design for each floater was introduced, with an emphasis on the design limits.

A preliminary verification of the FOWT models responses was done and the results were shown in section 6. We started by determining the static offset along with the natural frequencies. Afterwards, the controller's performance was tested using step wind simulations. Then, the effect of mean drift forces from second order waves was shown using regular waves simulations. Next we showed the effect of the overall difference second order wave forcing using irregular waves simulations. Finally, the dynamic response of the models were presented using different load cases with turbulent wind and irregular waves. Through



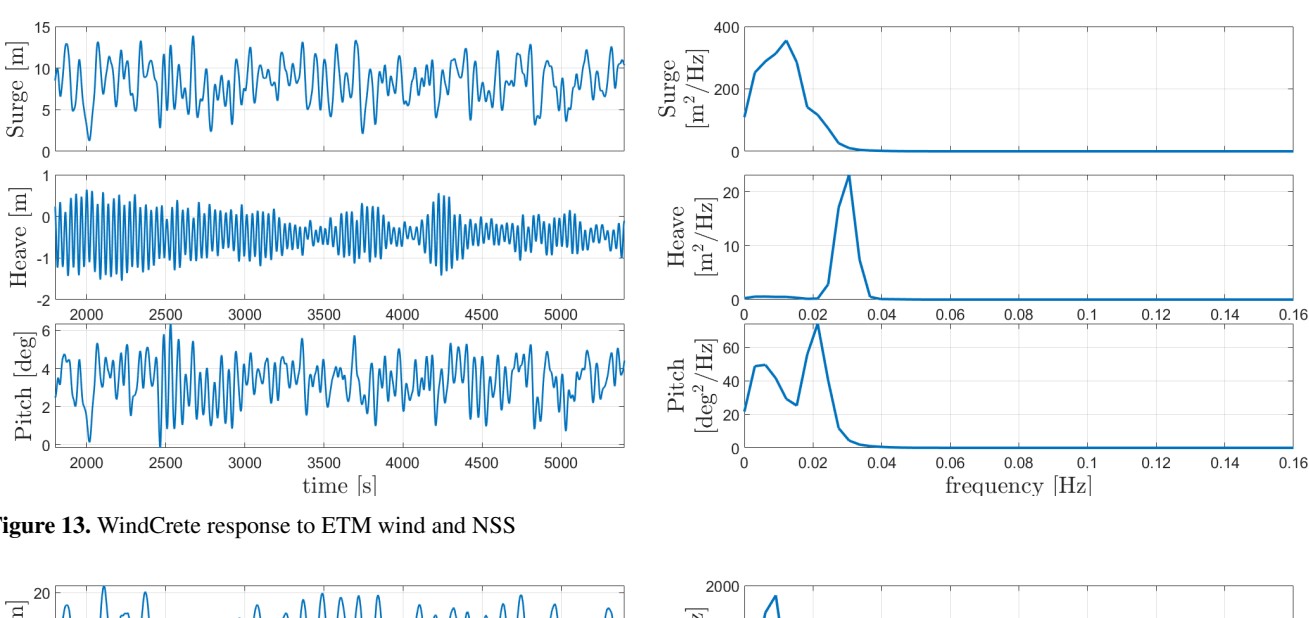

**Figure 13.** WindCrete response to ETM wind and NSS

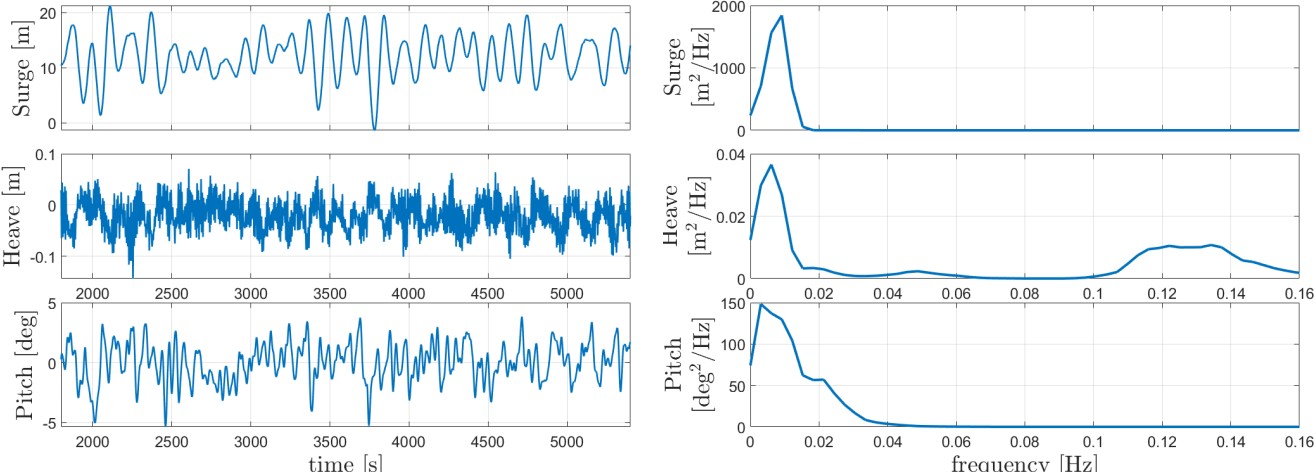

**Figure 14.** Activefloat response to ETM wind and NSS

the entire verification system we payed closer attention to the hydrodynamic motion responses of the floaters to understand the forces which dominate the motions response.

For the Gran Canaria site with mild wave loads, the motion responses were dominated by low frequency forces, at the natural frequencies of the floaters. The regular and irregular wave simulations' results showed that the second order waves played a significant role on the floater's rigid body motion responses. However, we emphasise that the low frequency resonance caused

by the second order waves is highly affected by the damping introduced in the hydrodynamic models. The damping currently introduced in the models came from literature. Experimental data is to be applied for tuning from wave tank tests. Afterwards, when the NTM wind field was simulated with the ESS irregular waves in Figures 11, and 12, the effect of the second order wave forces on the motions' frequency response became very small compared to the effect of the wind forces. Finally, in





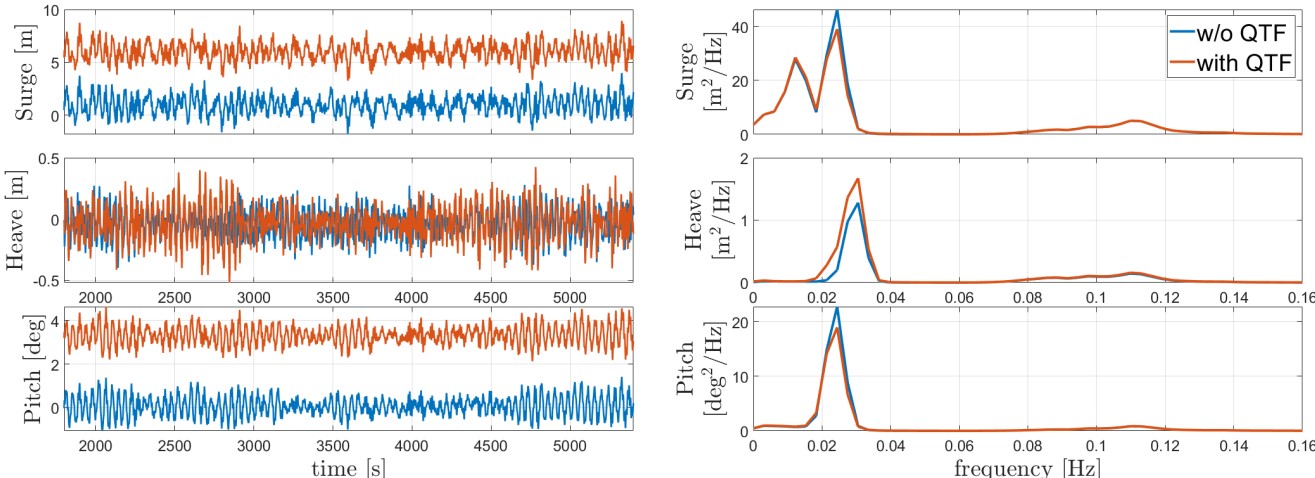

**Figure 15.** WindCrete response to EWM 50 years wind and ESS

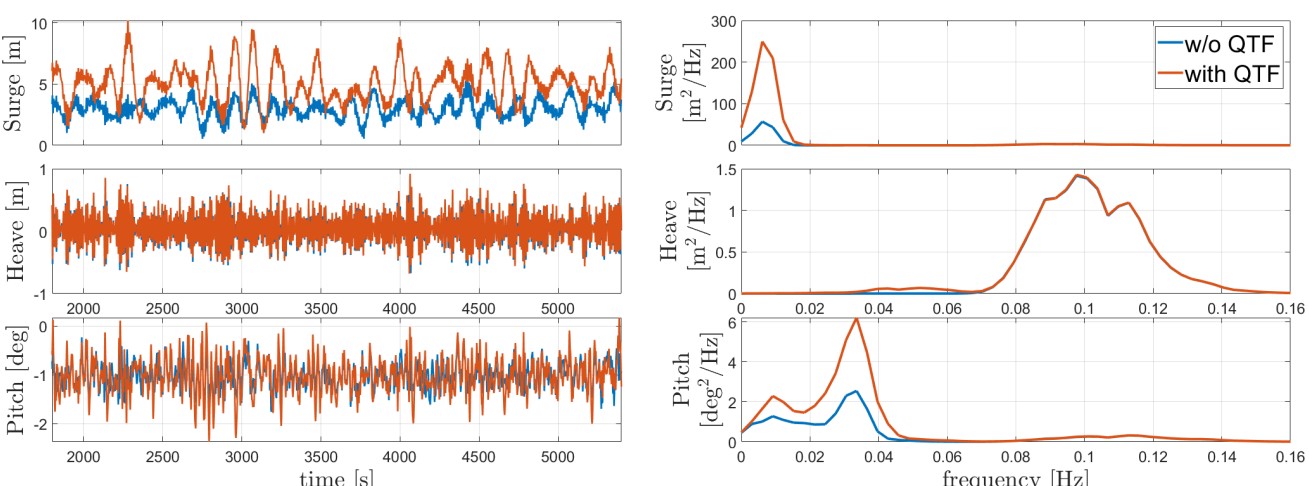

**Figure 16.** Activefloat response to EWM 50 years wind and ESS

all load cases the motions' responses were always dominated by the low frequency forcing. Therefore we conclude that the

models responses for the Gran Canaria site are mostly dominated by wind forces. The second order wave forces play a role in the motions' responses especially in surge, while the linear wave forces do not have a significant impact on the response of the system. This is due to the bigger size of the wind turbine, which increases the overall inertia of the system.



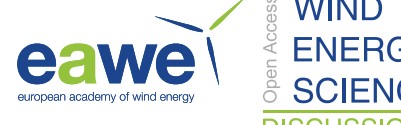

**Figure A1.** Activefloat response to step wind in absence of waves



## Appendix A: Activefloat response to step wind in absence of waves

*Author contributions.* M. Y. Mahfouz coupled Activefloat to the 15 MW OpenFAST model, created the potential flow model for WindCrete

in ANSYS-AQWA, and responsible for all the simulations done. C. Molins and P. Trubat designed the WindCrete floater and coupled it to OpenFAST. S. Hernández and F. Vigara designed the Activefloat floater. A. P. Jurado and H Bredmose, provided guidance and help in coupling the floaters to the OpenFAST models. M. Salari re-tuned the ROSCO controller to use for the floating platforms.

*Code and data availability.* The OpenFAST model of WindCrete and Activefloat are open access and can be found at Molins et al. (2020), and Duran et al. (2020). The results and data used to create all figures through the paper can be obtained by contacting the first author

*Competing interests.* The authors declare that they have no conflict of interest.

*Acknowledgements.* The research leading to these results has received partial funding from the European Union's Horizon 2020 research and innovation program under grant agreement No. 815083(COREWIND). I would like to thank my colleagues at Stuttgart Wind Energy (SWE) for reviewing this work.



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
