# Peer review of "Response of the IEA Wind 15 MW – WindCrete and Activefloat floating wind turbines to wind and second-order waves"

_Wind Energy Science, 2021_

## Referee Comment (RC1)

**General comments:**

Research objectives: "The goal of this work is to investigate the floaters responses at different load cases." Is a bit vague as a research objective. The design limits that where checked for should be more clearly highlighted.

As the authors stated this works is somewhat lacking of experimental validation, that will be performed in a following stage of the work. The very high wind excitation, around the floater's natural frequencies is acknowledged by the authors but the explanation of this phenomena can be improved. To address this, and also to benchmark the performance of the novel floater concepts, I would suggest to compare the response of the ActiFloat concept to the reference U-Maine floater design for the IEA 15MW made available by C. Allent et al. "Definition of the UMaine VolturnUS-S Reference Platform Developed for the IEA Wind 15-Megawatt Offshore Reference Wind Turbine". This would also allow for one to see how a more "standard" design fares in a site with mild sea conditions.

Results: A critical interpretation of the results shown would greatly help to illustrate advantages and disadvantages of the proposed concepts. Also the readability of this section would improve if the layout is changed, there are often more than 2 pages between figures and the point in the text where they are referenced.

**Specific Comments:**

Section 2.2: Please include the number and range of frequencies for which the potential flow problem is solved or reference to document where they can be found. This can be quite useful as a guideline for readers attempting to model similar systems.

Line 120: Hub-height is lower than IEA 15MW nominal value (150m). This has obvious benefits on stability as it lower COG and point of thrust application but may increase blade fatigue due to increased wind shear. Was this evaluated during design?

Although in the WindCrete concept tower and platform are a unique piece on concrete, these are modelled in OpenFAST as a flexible tower and rigid platform correct? Is this assumption reasonable? Please clarify.

Line 165: The way the active ballast system is implemented in OpenFAST is unclear. Is the floater's CG changed according to the values in table 6 based on the mean wind speed of the simulation?

Mafouz 2020 "Public design and FAST models of the two 15MW floater-turbine concepts"

Line 200: The loadcases used represent a standard set for verification. However when verifying "open" designs many authors include more extensive datasets that are often based on international standards (see Allen et al. Definition of the UMaine VolturnUS-S Reference Platform Developed for the IEA Wind 15-Megawatt Offshore Reference Wind Turbine). Please justify the choice of the loadcases in the present study.

FIGURES 3 & 4: For the surge decay of the windcrete and pitch decay of the activefloat concepts there seems to be a low-frequency component superimposed to the natural response frequency. What would be the cause of such phenomenon?

Line 240: What are the initial conditions in the model? Why weren't initial conditions imposed to be equal to the mean value reached during the simulation to shorten initial transients? I am afraid that results in regular waves may be quite inaccurate with such large initial transients compared to "stady state" response.

Figures 6&7: I am afraid that the initial transients will have a non-neglegible impacto on the PSDs shown.

Line 248, Figure 8: Pitch-surge couplings will be observed if the motion of the FOWT is not plotted with respect to it's istantaneous center of rotation. Decoupling these two DOFs is therefore only possible if motion is shown with respect to it's center of rotation. I believe that showing the motion with respect to the COG only introduces more confusion.

Lines 255-260: I think this part could be rephrased to improve clarity. The first time I read this section it seemed to me that the authors were implying some sort of connection between forcing introduced by QTFs and that introduced by Morrison Drag, which is not the case.

Line 263: "the Activefloat active ballast system is now activated to keep the mean static pitch of the platform around zero" could you clarify what this means? See previous comments.

Line 268: "In Figure 11, the frequency response shows a surge, pitch coupling." Can you explain this better?

Line 272: This is interesting. Wind is often thought of as a damping force. The fact that a NTM spectrum excites the platform's natural frequencies seems worrying to me. Can this be mitigated with controller tuning?

Conclusions: Impact and significance of this work should be highlighted more clearly. For instance the dominant wind-driven excitation of natural frequency observed with this large-thrust rotor is not well highlighted. Also a discussion on how this issue can be mitigated should be provided. Some of the statements in the conclusions, such as "For the Gran Canaria site with mild wave loads, the motion responses were dominated by low frequency forces, at the natural frequencies of the floaters." should also be stated in the results. This would provide this section with some much needed interpretation and not only description of what is shown in the plots.

---

## Referee Comment (RC2)

[referee-annotated manuscript omitted]

---

## Referee Comment (RC3)

[referee-annotated manuscript omitted]

---

## Author Comment (AC1)

**Authors' response to reviewer 1**

We thank the reviewer for the valuable comments and suggestions, which we consider very important and help us sharpen and improve the manuscript. Here are our responses to each comment.

The authors response is shown in green.

**General comments:**

Research objectives: "The goal of this work is to investigate the floaters responses at different load cases." Is a bit vague as a research objective. The design limits that were checked for should be more clearly highlighted.

As the authors stated this works is somewhat lacking of experimental validation, that will be performed in a following stage of the work. The very high wind excitation, around the floater's natural frequencies is acknowledged by the authors but the explanation of this phenomena can be improved. To address this, and also to benchmark the performance of the novel floater concepts, I would suggest to compare the response of the Activefloat concept to the reference U-Maine floater design for the IEA 15MW made available by C. Allent et al. "Definition of the UMaine VolturnUS-S Reference Platform Developed for the IEA Wind 15-Megawatt Offshore Reference Wind Turbine". This would also allow for one to see how a more "standard" design fares in a site with mild sea conditions.

Results: A critical interpretation of the results shown would greatly help to illustrate advantages and disadvantages of the proposed concepts. Also the readability of this section would improve if the layout is changed, there are often more than 2 pages between figures and the point in the textwhere they are referenced.

We have two objectives for our work. First, we want to introduce the floaters' designs and their coupling to the 15MW IE-Wind reference model. Second, we want to assess the floaters response at different load cases, with a focus on the effect of second order forces on low frequency response of the floater. We added a part to the introduction to make our objectives clearer. The designs were checked against the design limits presented in [1], which is now added to the introduction.

Comparing the two semi-submersible concept is really interesting, for better understanding of both designs and the effect of different design parameters on the system's response. However, comparing to other designs such as U-Maine is out of scope for our goals within this work for the following reasons. First, we are introducing the new floater designs as an open source reference models, to be further used within the research community for research purposes. Second we are presenting two different floater concepts, a spar and a semi-submersible. It will be unfair for the spar WindCrete model to shed more light on the semi-submersible design. Finally, COBRA the owner of the Activefloat

design are against direct comparison with different models in the work done within COREWIND.

We agree with the reviewer that the current position of the figures creates confusion. However, the layout of the figures will be adapted by the publisher in the last phase after the reviewings are done. This will increase the clarity and enhance the flow of the paper.

**Specific Comments:**

Section 2.2: Please include the number and range of frequencies for which the potential flow problem is solved or reference to document where they can be found. This can be quite useful as a guideline for readers attempting to model similar systems.

This information is added to the text.

Line 120: Hub-height is lower than IEA 15MW nominal value (150m). This has obvious benefits on stability as it lower COG and point of thrust application but may increase blade fatigue due to increased wind shear. Was this evaluated during design?

This is a very interesting remark. Our focus is on the novel floater design and the combination with the IEA Wind 15 MW. Hence we think that investigations like the one suggested by the reviewer will be part of future work.

Although in the WindCrete concept tower and platform are a unique piece on concrete, these are modelled in OpenFAST as a flexible tower and rigid platform correct? Is this assumption reasonable? Please clarify.

Yes, it is correct. The tower is modeled as a flexible body and the floater is considered rigid following standard modelling procedures in FAST, the floater below MWL was modelled as rigid. Since the floater diameter and wall thickness are larger than for the tower below MWL, this approach can be regarded as a good first approximation. An assessment of the effect of floater flexibility for a 10 MW spar floater was presented in [2]. It was found, for that floater, that when a flexible mode was introduced, the overall response behaviour of the rigid body modes were preserved although with some increase in the response amplitude operators to linear wave forcing of surge and pitch. The largest increase was of 17% and occurred at the natural frequencies of the surge and pitch modes

Moreover, The division of the tower and the platform in two different parts in the numerical model, does not affect the monolithic concept of WindCrete. When we check WindCrete's design limit against ultimate and fatigue loads over the tower height, the Minimum Breaking Load (MBL) is defined only by the MBL of the concrete material, and no connection point is assumed in the structure.

Line 165: The way the active ballast system is implemented in OpenFAST is unclear. Is the floater's CG changed according to the values in table 6 based

on the mean wind speed of the simulation? Mahfouz 2020 "Public design and FAST models of the two 15MW floater-turbine concepts"

This information is added to the text.

Line 200: The loadcases used represent a standard set for verification. However when verifying "open" designs many authors include more extensive datasets that are often based on international standards (see Allen et al. Definition of the UMaine VolturnUS-S Reference Platform Developed for the IEA Wind 15-Megawatt Offshore Reference Wind Turbine). Please justify the choice of the loadcases in the present study.

We believe we misused the word verification through the paper. The main goal of the paper is not the verification of the models. The main goal is to present the floaters to the research community and to analyze and assess the floaters' performance at different load cases with an emphasis on the second order wave forces effects. We believe the set of load cases we used achieved this goal, and showed the behaviour of both floaters.

Figures 3 and 4: For the surge decay of the windcrete and pitch decay of the activefloat concepts there seems to be a low-frequency component superimposed to the natural response frequency. What would be the cause of such phenomenon?

Yes, this is correct. We tried to explain this by the sentence in line 228 "The surge decay includes not only one frequency, but a combination of different frequencies because it is measured at the mean sea level and not at the CG of the FOWT system.". We added to this sentence now to clarify.

Line 240: What are the initial conditions in the model? Why weren't initial conditions imposed to be equal to the mean value reached during the simulation to shorten initial transients? I am afraid that results in regular waves may be quite inaccurate with such large initial transients compared to "steady state" response. Figures 6 and 7: I am afraid that the initial transients will have a non-neglegible impact on the PSDs shown.

For all the simulations with wind fields, the initial conditions used are corresponding to the platform mean equilibrium position when a steady wind field is acting on the turbine in the absence of waves.

The regular waves simulations were done without setting the correct initial conditions, hence the WindCrete platform does not reach steady state. This load case is now repeated with the correct steady states to decrease the transient time, and the platform reaches steady state in less than 1500 s. For the PSDs drawn in Figures 6 and 7, we excluded the first 1500 s, the figures shown only analyse the frequency response of the last 1500 s. We added this information to the text.

Line 248, Figure 8: Pitch-surge couplings will be observed if the motion of the FOWT is not plotted with respect to it's instantaneous center of rotation.

Decoupling these two DOFs is therefore only possible if motion is shown with respect to it's center of rotation. I believe that showing the motion with respect to the COG only introduces more confusion.

Yes, that's correct. However, we think that using the CG is enough for the following reasons. First, the difference between the CG and the instantaneous center of rotation is very small. Second, we only wanted to show that second order forces have a limited effect on the surge motion and to prove that the big difference in the surge response shown in Figure 6 is only because the surge motion is measured at the sea water level. If we shift the measuring point to the CG as in Figure 8, it is clear that only pitch DOF is highly affected by the second order wave forces. We can see now that our explanation was not clear and we rephrased that part.

Lines 255-260: I think this part could be rephrased to improve clarity. The first time I read this section it seemed to me that the authors were implying some sort of connection between forcing introduced by QTFs and that introduced by Morrison Drag, which is not the case.

This is modified now in the text.

Line 263: "the Activefloat active ballast system is now activated to keep the mean static pitch of the platform around zero" could you clarify what this means? See previous comments.

This is clarified in the answer to a previous comment

Line 268: "In Figure 11, the frequency response shows a surge, pitch coupling." Can you explain this better?

This is now added to the text. Vertical lines showing the natural frequencies of different DOFs are also now added to the images to provide a clearer illustration.

Line 272: This is interesting. Wind is often thought of as a damping force. The fact that a NTM spectrum excites the platform's natural frequencies seems worrying to me. Can this be mitigated with controller tuning?

A similar response at operation conditions with NTM wind field was found for the 10MW DTU reference model coupled to OO-Star Wind Floater, which can be seen in [3]. We agree that this is an interesting effect to keep in mind. However, as long as the forces and the excitation limits are within the design specification for different load cases, this is a normal response especially due to the higher thrust forces in the 15MW reference model. Further tuning of the baseline controller used will have a small effect on these responses, an advanced controller approach with extra loops to account for the platform accelerations could be applied to damp the platform's responses.This is part of future research focusing on optimizing the models.

Conclusions: Impact and significance of this work should be highlighted more

clearly. For instance the dominant wind-driven excitation of natural frequency observed with this large-thrust rotor is not well highlighted. Also a discussion on how this issue can be mitigated should be provided. Some of the statements in the conclusions, such as "For the Gran Canaria site with mild wave loads, the motion responses were dominated by low frequency forces, at the natural frequencies of the floaters." should also be stated in the results. This would provide this section with some much needed interpretation and not only description of what is shown in the plots.

Thank you for these suggestions. The conclusions are updated now to include them.

**References**

[1] Fernando Vigara, Lara Cerdán, Rubén Durán, Sara Muñoz, Mattias Lynch, Siobhan Doole, Climent Molins, Pau Trubat, and Raúl Gunache. Design basis, September 2020.

[2] Michael Borg, Anders Melchior Hansen, and Henrik Bredmose. Floating substructure flexibility of large-volume 10MW offshore wind turbine platforms in dynamic calculations. *Journal of Physics: Conference Series*, 753(8), 2016.

[3] Antonio Pegalajar-Jurado, Henrik Bredmose, Michael Borg, Jonas G. Straume, Trond Landbø, Hakon S. Andersen, Wei Yu, Kolja Müller, and Frank Lemmer. State-of-the-art model for the LIFES50+ OO-Star Wind Floater Semi 10MW floating wind turbine. *Journal of Physics: Conference Series*, 1104(1), 2018.

---

## Author Comment (AC2)

**Authors' response to reviewer 2**

We thank the reviewer for the valuable comments and suggestions, which we consider very important and help us sharpen and improve the manuscript. Here are our responses to each comment.

The authors response is shown in green.

**General comments:**

Table 1: Title center
The titles' format are following the publisher's tex template.

Could you please provide some more details as to how the controller was tuned, in case somebody would like to reproduce these results. Or can the controller also be released to the public?
Thanks for your question. A reference describing the controller tuning in details is added to the text. The source code of the ROSCO controller itself is open access and can be found at [1]. We did not do any changes to the controller's source code. We only tuned the controller gains, which can be found in the ServoDyn file in the OpenFAST models.

Suggest plotting different line styles (dashed) to be able to differentiate and/or show that they overlap exactly. Otherwise you literally don't see the plot w/o QTF
All figures are updated following this suggestion.

For these plots and almost all other plots, it would be extremely helpful to have the natural frequency of the floater and, when appropriate the wave forcing frequencies in the plots also (as vertical lines simply). This allow instant recognition and supports the statements made in the text.
All figures are updated following this suggestion.

All minor changes for spelling mistakes and structure are now implemented in the text.

**References**

[1] NREL. ROSCO. Version 1.0.0, 2020.

---

## Author Comment (AC3)

**Authors' response to reviewer 3**

We thank the reviewer for the valuable comments and suggestions, which we consider very important and help us sharpen and improve the manuscript. Here are our responses to each comment.

The authors response is shown in green.

**General comments:**

Line 11: What about for large waves? What are the conditions that produce the largest loads?

Only the environmental conditions of the Gran Canaria site are used in the paper. The Gran Canaria site has a mild environmental condition with fifty years extreme waves with $H_s = 5.11$m. Therefore we stated through the paper that this response is at mild sea conditions.

Line 12: Are you saying the models are now verified? What do you mean by verified?

Line 203: Again, what do you mean by verification? Do you simply mean assessment or investigation?

Line 300: I would not use the term verification. Either assessment or investigation. Verification is defined as determining that there is either no error in the modeling theory implementation, or that your simulation result is adequately converged. That is not the focus here.

Thanks a lot for this feedback. We agree that this is a misuse of the word verification and we have modified the paper according to your suggestion.

Line 21: Relative to what baseline?

Thanks for this comment. It is now clarified in the text

Line 55: New version of OpenFAST now allows for flexible substructure (platform) as well.

We are aware that NREL are updating SubDyn to model flexible floaters. However, to the best of our knowledge it is not publicly available yet.

Line 99: Aren't your tower natural frequencies above the linear wave-excitation region?

Thanks for the comment, this is a very interesting question. The sum QTF may be able to excite the first coupled tower frequency, this is expectedly by a small amplitude, since wave-driven excitation of the tower will have to happen through motion-excitation of the floater, which is modelled as a rigid structure. This is supported by the findings [1]. The explanation is added at the end of section 2.2.

Moreover, the tower loads will always be dominated by the wind loading especially due to the higher thrust forces on the 15 MW reference turbine, which further supports the neglection of the sum frequency wave forcing.

Table 9: For Activefloat, can you explain how your static equilibrium has a large negative pitch, but positive surge?

Static equilibrium is found with the mooring system attached. Both floaters have a negative pitch due to the big overhang of the tower top, causing a negative moment. In the absence of wind and waves, the surge motion is only affected by the mooring line forces, having a positive surge means that the mooring lines are pulling the platform in the positive $x$ direction. This explanation is now added to the paper.

Table 10: Is there concern with the coalescence of the heave natural frequency for Activefloat with waves?

The heave natural period is much higher than the peak wave period $(T_p)$ in the Gran Canaria islands which is between 6 s at normal operations and 9 s at extreme conditions, hence the heave natural frequency lies outside of the wave frequency range. We have not seen any concerning responses in any load case, and we believe there is no coalescence. Figure 1 shows Activefloat's heave natural frequency with the vertical line, and the wave frequency range for $T_p = 9s$.

[Figure]

Figure 1: Irregular wave frequency range $T_p = 9s$

Line 233: I'm not sure one would encounter negative damping during a step wind event, but rather during operation as the turbine is oscillating in surge/pitch and the controller is reacting to the oscillating perceived wind speed.

The steps in a step test induce a change in thrust and will induce a transient settlement into the new equilibrium. The step test is therefore a good first check that negative damping does not happen due to sudden changes (steps) in the wind speed. The purpose here is to check that negative damping does not happen at any wind speed, and not during the ramp from one wind speed to the other. Therefore, after each step in the wind speed, the steady wind field for 200 s is used to check that the controller does not add any negative damping to the system before stepping again to a higher/lower wind speed. If the controller is not tuned the negative damping effect can still happen with steady wind fields. Since we can not see any resonance or negative damping in both the step wind simulation and also in all the other load cases with turbulent wind field, we came up with the conclusion that the controller does not add negative damping to the system. We updated the manuscript to include this statement.

Line 239: And without wind. The wind forces may override the wave-drift forces when present.
Thanks for the comment. The text is now updated.

If after 3000 seconds, the WindCrete is still not in equilibrium for regular waves, is 1800 seconds a sufficient amount of time for transient removal?
The regular waves simulations were done without setting the correct initial conditions, hence the WindCrete platform does not reach steady state. This is now corrected and the load case is redone to decrease the transient time, and the platform reaches steady state in less than 1800s. For all the other simulations with wind fields, the initial conditions used are corresponding to the platform equilibrium position when a steady wind field is acting on the turbine.
In Figure 9 and 10 for the response to irregular waves in the absence of a wind field, we use the same starting initial conditions for both load cases and run the simulation for 5400s. This is enough for the purpose of comparing the responses of the platforms to irregular waves with and without second order forces. Moreover, we believe the effect of the transients on the motion response is minimum when compared to the first and second order wave forces, hence removing the first 1800s is enough to give credible results.

Line 260: and since the natural frequency lies in the linear wave excitation region.
We updated the text to this "and since the natural frequency lies close to the wave excitation region" as we have shown earlier that the heave frequency is close to the wave excitation but it doesn't lie within the wave frequency range.

Would be useful to state the wave properties in each figure caption.
The captions are now updated to include wave properties.

All minor changes for spelling mistakes and structure are now implemented in the text.

**References**

[1] Sébastien Gueydon, Tiago Duarte, and Jason Jonkman. Comparison of Second-Order Loads on a Semisubmersible Floating Wind Turbine. In *Volume 9A: Ocean Renewable Energy*, number 5, pages 1–12. American Society of Mechanical Engineers, jun 2014.